# Synthesis of Zeolites from Fine-Grained Perlite and Their Application as Sorbents

**DOI:** 10.3390/ma15134474

**Published:** 2022-06-24

**Authors:** Florian Painer, Andre Baldermann, Florian Gallien, Stefanie Eichinger, Florian Steindl, Reiner Dohrmann, Martin Dietzel

**Affiliations:** 1Institute of Applied Geosciences & NAWI Graz Geocenter, Graz University of Technology, Rechbauerstraße 12, 8010 Graz, Austria; baldermann@tugraz.at (A.B.); stefanie.eichinger@tugraz.at (S.E.); florian.steindl@tugraz.at (F.S.); martin.dietzel@tugraz.at (M.D.); 2Omya GmbH, Gersheimstrasse 1-2, 9722 Gummern, Austria; florian.gallien@omya.com; 3Federal Institute for Geosciences and Natural Resources (BGR), Stilleweg 2, 30655 Hanover, Germany; reiner.dohrmann@lbeg.niedersachsen.de; 4State Authority of Mining, Energy and Geology (LBEG), Stilleweg 2, 30655 Hanover, Germany

**Keywords:** LTA-zeolite, zeolite synthesis, perlite fines, heavy metals, wastewater treatment

## Abstract

The hydrothermal alteration of perlite into zeolites was studied using a two-step approach. Firstly, perlite powder was transformed into Na-P1 (GIS) or hydro(xy)sodalite (SOD) zeolites at 100 °C and 24 h using 2 or 5 M NaOH solutions. Secondly, the Si:Al molar ratio of the reacted Si-rich solution was adjusted to 1 by Na-aluminate addition to produce zeolite A (LTA) at 65 or 95 °C and 6 or 24 h at an efficiency of 90 ± 9% for Al and 93 ± 6% for Si conversion. The performance of these zeolites for metal ion removal and water softening applications was assessed by sorption experiments using an artificial waste solution containing 4 mmol/L of metal ions (Me^2+^: Ca^2+^, Mg^2+^, Ba^2+^ and Zn^2+^) and local tap water (2.1 mmol/L Ca^2+^ and 0.6 mmol/L Mg^2+^) at 25 °C. The removal capacity of the LTA-zeolite ranged from 2.69 to 2.86 mmol/g for Me^2+^ (=240–275 mg/g), which is similar to commercial zeolite A (2.73 mmol/g) and GIS-zeolite (2.69 mmol/g), and significantly higher compared to the perlite powder (0.56 mmol/g) and SOD-zeolite (0.88 mmol/g). The best-performing LTA-zeolite removed 99.8% Ca^2+^ and 93.4% Mg^2+^ from tap water. Our results demonstrate the applicability of the LTA-zeolites from perlite for water treatment and softening applications.

## 1. Introduction

Anthropogenic activities, such as mining, transportation, agriculture, households, industry and urbanization, have led to a significant accumulation of environmentally critical heavy metal ions (Me: Cd^2+^, Cr^3+^, Cu^2+^, Fe^2+^, Mn^2+^, Ni^2+^, Pb^2+^, Zn^2+^ etc.) in most of the aquatic and terrestrial environments of the Earth’s surface [1,2,3,4]. These Me ions can be hazardous and health-damaging for human beings and for the ecological system, causing diseases and disorders in living organisms even at low concentrations. Therefore, advanced solutions to decrease the Me ion concentration below permissible limits, defined by local and international standards, are required [5,6,7]. Current methods to treat water contaminated by Me ions include, for instance, chemical oxidation and precipitation, flocculation/sedimentation, electrochemical methods, membrane separation and (ad)sorption, but these wastewater treatment technologies vary greatly in environmental compatibility, effectiveness, operational costs, materials availability, and sustainability [8,9]. Thus, there is a growing demand to develop novel or to tailor existing materials with appropriate physicochemical properties (i.e., high surface area, small particle size and high affinity to bind Me ions) that can be used for the efficient, green, sustainable, and low-cost treatment and conditioning of wastewater [10,11,12,13,14].

Zeolites are proper candidate minerals for wastewater treatment and conditioning, including use for the reduction of water hardness and the removal of diverse Me ions from solution, owing to their small particle size, high surface area and uniform pore structure [15,16]. Moreover, zeolites have received increasing attention for the remediation of wastewater contaminated by a suite of harmful and potentially radioactive Me ions, such as Cd^2+^, Co^2+^, Cr^3+^, Cs^+^, Pb^2+^, Sr^2+^ and Zn^2+^, due to their high thermal and radiation stability and their worldwide abundance [5]. SOD- and GIS-zeolites have also exhibited a superb performance in, e.g., the pervaporative desalination of seawater, Pb^2+^ removal from solution and wastewater treatment [17,18,19]. Furthermore, synthetic zeolite A (LTA) was found to be highly efficient in the treatment of wastewater contaminated with Me ions [20,21,22,23] and in water softening [24,25,26].

The synthesis of high-quality LTA-zeolite as well as other commercially available zeolites, is often done with expensive chemical grade silicon (colloidal silica, precipitated silica or fumed silica) and alumina (gibbsite, aluminate salts or metal powder) compounds [27,28]. Alternative synthesis routes for zeolite production may also use a variety of natural raw materials of comparatively low price, including kaolinite [29,30], pumice [31], diatomite [32] and fly ash [19,33], while the use of natural reagents over commercial ones has economic benefits, the complex chemical composition of natural materials increases the risk of zeolitic products formation with variable composition and purity [34]. For these reasons, the production of, e.g., LTA-zeolites for commercial water treatment applications is still limited caused by (i) the complexity of the synthesis process, (ii) the involvement of seeds of high costs in the synthesis, (iii) difficulties to obtain reproducible and defect-free (pure) products, (iv) the unsuitable performance of most zeolites under acidic conditions, (v) the fragile structure and the corresponding low mechanical stability of most zeolites under corrosive conditions and (vi) technical challenges to manufacture porous zeolite membranes [28]. In essence, while the use of natural zeolites in water treatment is cost-efficient, the coefficient of performance is often lower compared to the application of synthetic, high-performance LTA-zeolites at industrial scale. For this reason, the production of LTA-zeolites from industrial raw materials can serve as a “green solution”, as this approach reduces the demand for primary raw materials and minimizes the costs for residues disposal [35].

In the literature, fine-grained natural perlite, expanded perlite and (expanded) perlite waste are used as raw materials for the synthesis of zeolites as well (e.g., [34,36,37]). Two possible ways for the zeolitization from perlite are commonly distinguished: (i) direct zeolite precipitation under hydrothermal conditions without chemical pre-treatment of the starting material to yield the target zeolite, usually in addition to variable amounts of inert crystalline impurities, such as quartz, feldspar, and mica [34,36,38]. (ii) A two-step reaction process is used, where perlite is hydrothermally treated with a NaOH solution to typically obtain non-target zeolites and additional solid-phase impurities in a first reaction step. In a second step, the solid reaction product(s) are removed by filtration, the Na:Si:Al molar ratios of the remaining solution is chemically readjusted and exposed to hydrothermal treatment to yield the target zeolite of high purity [37,39].

In this study, a two-step process was adapted to transform industrial-grade perlite powder within a recycling economy approach to obtain high-purity and high-quality LTA-zeolites at a high degree of efficiency (judged from the total loss of dissolved Al and Si from solution after each reaction step, zeolite purity and product yields). Another focus was the synthesis of a high added value product after the initial treatment step and thus preventing material loss. An overview of zeolitic framework structures and compositions relevant for this study are summarized in Table 1 and Figure 1. The synthesized zeolites were evaluated for their potential (i) to immobilize Me ions from a multi-component waste solution and (ii) to reduce the hardness of typical freshwater (local tap water) by running a set of laboratory-scale batch experiments.

## 2. Materials, Experimental and Methods

### 2.1. Materials

Industrial-grade perlite powder is a non-expandable, fine-grained by-product of limited use that accumulates during the processing of raw perlite ore. The perlite powder studied here was used as received without any pre-treatment. The material is light grey in color, has a grain size of <55 µm with a median diameter (d50) of 17 µm (Figure 2A), and a specific surface area (SSA) of 2.9 m^2^/g. The molar Si:Al ratio (SAR) is 4.33; the chemical composition is indicative of a rhyolitic chemistry (Table 2). The mineralogical composition is dominated by an amorphous phase (∼96.0 wt.%) and minor quartz (2.0 wt.%), feldspar (1.5 wt.%), and muscovite (0.5 wt.%) (Figure 2B). The infrared spectroscopic analysis (Figure 2C) reveals a broad adsorption at ∼1000 cm^−1^ and a smooth one at ∼1160 cm^−1^, which belong to the asymmetric stretching vibrations of Si-O-Si and Si-O-Al planes [41]. Adsorption in the region from 1250 to 850 cm^−1^ is characteristic for an amorphous alumosilicate network and the weak adsorption at 1630 cm^−1^ is related to the deformation mode of molecular water in raw perlite [41]. The electron microscopic inspection of the perlite shows the amorphous phase to be composed of volcanic glass shards (Figure 2D).

Besides natural perlite powder, which provides a silica source, technical anhydrous sodium aluminate (NaAlO_2_, ≥99.95%, Sigma-Aldrich, St. Louis, MO, USA) was added as an alumina source to adjust the Si:Al ratio during zeolite synthesis. Solid sodium hydroxide pellets (NaOH, ≥97%, Sigma-Aldrich) were dissolved in ultrapure water to obtain 2 and 5 M NaOH solutions, which were used as a mineralizing agent in the zeolite extraction and synthesis experiments. For the selective ion exchange experiments, adequate amounts of calcium chloride dihydrate (CaCl_2_·2H_2_O, ≥99%, Carl Roth, Karlsruhe, Germany), magnesium chloride hexahydrate (MgCl_2_·6H_2_O, ≥99%, Carl Roth), barium chloride dihydrate (BaCl_2_·2H_2_O, ≥99%, Carl Roth) and zinc chloride (ZnCl_2_, ≥97%, Carl Roth) were dissolved in ultrapure water to obtain a synthetic solution containing equimolar (∼4 mmol/L) concentrations of Ca^2+^, Mg^2+^, Ba^2+^ and Zn^2+^ ions. The pH value of the Me stock solution was set to 6.0 ± 0.1 by the dropwise addition of a 0.1 M hydrogen chloride solution (HCl, p.a., Sigma-Aldrich) under continuous stirring. Graz tap water was used in the water softening experiments, which contained the following cationic concentrations: 2.1 mmol/L Ca^2+^, 0.6 mmol/L Mg^2+^, 0.3 mmol/L Na^+^ and 0.03 mmol/L K^+^ at a pH value of 7.8 ± 0.03. A commercial LTA-zeolite (UOP, Honeywell Company, Charlotte, NC, USA) was used as a reference material for the subsequent evaluation of the performance tests. All experimental solutions were prepared with ultrapure water (Milli-Q Plus UV, Millipore, Burlington, MA, USA, 18.2 MΩ at 25 °C).

### 2.2. Zeolite Synthesis

A two-step reaction process for the hydrothermal synthesis of zeolites from perlite was adapted (Figure 3; [37,39,42]) to obtain selected zeolite frameworks with defined compositions and physicochemical properties in each reaction step. The synthesis conditions were adapted to yield SOD-, GIS- (“extraction” step) and especially LTA-zeolite (“synthesis” step), which are well-known to exhibit a (very) high cation exchange capacity (CEC) and selective Me ion removal potential [29].

Consequently, 3 g of perlite raw material were reacted with 30 mL of 2 M or 5 M NaOH solutions at 100 °C for 24 h to obtain zeolites with a GIS or SOD framework structure (product E in Figure 3). All experiments were carried out in stainless steel autoclaves with a Teflon liner sealed with a Teflon cap. The autoclaves were placed in an oven equipped with a rotating overhead system to ensure complete homogenization of the suspensions.

After this first extraction step, the autoclaves were cooled down to room temperature within ∼1 h. Subsequently, the solids were separated by a vacuum filtration unit using cellulose acetate filters (0.45 µm pore size, Sartorius, Göttingen, Germany). The reacted extraction solutions were immediately filtered and subsequently used in the second reaction step. An aliquot of the extraction solution was diluted and acidified for analysis of the Na, Si and Al concentrations. The solid products obtained were washed several times with deionized water to remove electrolytes until the electric conductivity of the filtrate was determined to be <300 µS/cm. The remaining solids were dried in an oven at 40 °C for 24 h.

For reaction step 2, 15 mL of the isolated Si-rich filtrate obtained from extraction step 1 were mingled with solid NaAlO_2_ to instantaneously form a Na-alumosilicate gel with a SAR of 1.0 ± 0.2. The gel-like suspensions were homogenized using a magnetic stirrer (250 rpm for 5 min) before they were reacted in autoclaves at 65 °C or 95 °C for 6 h or 24 h, respectively, to obtain a target LTA-zeolite of high-purity (product S in Figure 3). At the end of these synthesis experiments, the reacted solutions and the solids were treated in the same way as described before.

An overview about the hydrothermal conditions used for the directed precipitation of zeolites in the extraction and synthesis steps is provided in Table 3 and Table 4. The efficiency of the zeolite synthesis was evaluated qualitatively by means of the purity of the reaction products E and S, and quantitatively by means of the fraction of Al and Si remaining in solution after the extraction step 1 versus the removal of Al and Si from the solution after the synthesis step 2 (see Table 3 and Table 4), according to the expressions: (1)$$\varepsilon =\frac{\Delta {\mathrm{Al}}_{start-end}}{\Delta {\mathrm{Al}}_{start}}\cdot 100$$
or
(2)$$\varepsilon =\frac{\Delta {\mathrm{Si}}_{start-end}}{\Delta {\mathrm{Si}}_{start}}\cdot 100$$
where *ε* is the removal efficiency of Al and Si (in%) and ΔAl and ΔSi denote the difference between the dissolved Al and Si concentrations measured at the start and end of the synthesis experiments.

### 2.3. Performance Tests

The performance of the synthesized zeolites was evaluated by means of quantifying and assessing the selective removal potential for different Me ions from an artificial multi-component Me stock solution and from Graz tap water (see Section 2.1). For the Me ion removal experiments, 50 mL of the Me-containing solutions were mingled with 0.1 g of zeolite in 50 mL high-density polypropylene (PP) vials to obtain a liquid–solid ratio of 500:1. The suspensions were placed on a horizontal shaker (Edmund Bühler KS-15B) for 24 h at 25 °C. Measurements of pH, EC and solution sampling were done at the beginning and the end of performance tests which lasted 24 h to ensure chemical equilibrium. The liquid samples were filtered using cellulose acetate syringe filters (0.45 µm pore size, BIOFIL) and prepared (i.e., diluted and acidified) for subsequent chemical analysis.

From these chemical data sets, the efficiency of the Me ion removal process was quantified according to the expressions: (3)$$q_{e}=\frac{C_{0}-C_{e}}{m}\cdot V$$
and
(4)$$\%removal=\frac{C_{0}-C_{e}}{C_{0}}\cdot 100$$
where $C_{0}$ and $C_{e}$ refer to the initial and final (sorption equilibrium) concentrations of the adsorbate (in mg/L), *m* is the dry mass of the adsorbent (in g) and *V* is the volume of the solution (in L). The reproducibility of all results was determined in triplicate. The relative standard deviations were determined to be always below ±3%. In the following, only the average values are reported. All experimental solutions were, at any time, undersaturated with respect to Me(II)-(oxy)hydrate, -carbonate, -chloride or -sulfate forms, as indicated by hydrochemical modelling of saturation indices (SI) of relevant mineral phases using the PHREEQC software and its implemented LLNL-database. The calculation also showed that the predominance of the dissolved Me ions prevailed in monovalent (Na^+^ and K^+^) or bivalent (Ca^2+^, Mg^2+^, Ba^2+^ and Zn^2+^) form throughout all experiments.

### 2.4. Analytical Methods

#### 2.4.1. Fluid-Phase Characterization

The pH value of the experimental solutions was measured at 25 °C with a BlueLine 28 electrode connected to a WTW pH/Cond 3320 pH-meter (Weilheim in Oberbayern, Germany ), which was calibrated against NIST buffer solutions at pH 4.01, 7.00, and 10.00 (analytical error: ±0.05 pH units).

A PerkinElmer Optima 8300 inductively coupled plasma optical emission spectrometer (ICP-OES, Waltham, MA, USA) was used to measure the concentrations of dissolved Al, Fe, K, Na, Ca, Mg, Ba, Zn and Si in acidified samples (2% HNO_3_ matrix), which were obtained from the extraction and synthesis solutions and from the performance tests. The analytical precision is better than ±4% for all dissolved components as determined based on replicate measurements of NIST 1640a, in-house and SPS-SW2 Batch 130 standards [14].

The aqueous concentrations of Na, K, Mg and Ca were measured by ion chromatography (Dionex ICS 3000, IonPac, Waltham, MA, USA) for the practical application experiments with Graz tap water.

#### 2.4.2. Solid-Phase Characterization

The mineralogy of the materials was examined by a PANalytical X’Pert PRO diffractometer (Malvern, UK) using Co–Kα-sourced radiation (λ = 1.79 nm) produced at 40 kV and 40 mA. The diffractometer is equipped with (analytical uncertainty: ≤5 wt.% [43]).

Fourier-transform infrared spectroscopy (FTIR) was carried out on a PerkinElmer Frontier device using the attenuated total reflectance configuration (ATR) to further identify the type and nature of the poorly crystalline educts and products. The spectra were collected in the mid-infrared (MIR) range (4000–650 cm^−1^) at a resolution of ±2 cm^−1^.

The crystal morphology (shape and size) of the samples was analyzed in a ZeissGemini DSM 982 field emission scanning electron microscope (SEM, Oberkochen, Germany) under high vacuum conditions at the University of Graz. Therefore, representative samples were mounted on Al-stubs, fixed with a C-film, and sputtered with Au/Pd alloy using a Scancoat Six sputter coater (Edwards Hochvakuum GmbH, Butzbach, Germany). Secondary electron images (SEI) were obtained using an acceleration voltage of 2 kV and a working distance of 5 mm.

Wavelength-dispersive X-ray fluorescence spectroscopy (XRF), conducted on a Philips PW 2404 machine (Amsterdam, The Netherlands), was used to determine the chemical composition of the perlite. The loss on ignition (LOI) was determined by calcinating 1.8 g of the pre-dried (at 105 °C) sample at 950 °C for 1 h and measuring the residual mass by gravimetric analyses. Then, 1 g of the calcined sample was mixed with 6 g of the fusion agent Li_2_B_4_O_7_. The mixture was homogenized, transferred into a platinum crucible, and fused in a bead preparation apparatus (Perl’X, PANalytical) at 1200 °C for 12 min. Data evaluation was performed using the software IQ+ (PANalytical). The analytical error was determined as ±0.5 wt.% for the major elements [44].

The specific surface area was calculated according to the BET method based on five-point nitrogen adsorption measurements using a Micromeritics surface analyzer VII 2390 (Norcross, GA, USA) and 200–400 mg sample mass. The sample was preconditioned by heating at 105 °C under vacuum for 24 h.

## 3. Results and Discussion

### 3.1. Hydrothermal Extraction Step

The XRD patterns of the solids obtained after the extraction step indicate the formation of a well-crystallized Na-P1 (GIS) (Figure 4A) or hydro(xy)sodalite (SOD) (Figure 4B) depending on the NaOH concentration (2 M or 5 M) used for zeolite synthesis, which is consistent with analogous observations made by Christidis et al. [34]. Minor amounts of quartz and muscovite, which are present in all solids, likely represent unreacted mineral phase impurities originating from the perlite raw material (cf. Figure 2B and Table 3).

The FTIR spectra obtained from the GIS- (Figure 4C) and SOD-zeolites (Figure 4D) show strong adsorption in the range from 970 to 940 cm^−1^, which is related to intra-tetrahedral Si-O-Si and Si-O-Al asymmetric stretching vibrations in zeolites [45]. The shoulder located at 1250–1050 cm^−1^ is attributable to the asymmetric stretching mode of external linkages between TO_4_ units in (alumo)silicate networks [46]. The position of the T-O-T asymmetric stretching band is highly sensitive to the Al content in a given zeolite framework [45,47,48]. The IR bands in the range of 800 cm^−1^ and 650 cm^−1^ correspond to the pseudolattice vibrations in zeolitic structures [48]. The peaks at about 739 cm^−1^, 675 cm^−1^ (Figure 4C) and 719 cm^−1^, 694 cm^−1^, 660 cm^−1^ (Figure 4D) are characteristic for the synthesized GIS- and SOD-zeolites, respectively, [45,49]. Adsorption in the range of 3800–3000 cm^−1^ (data are not shown) belong to the stretching vibrations of H-O-H (zeolitic H_2_O), SiO-H groups and internal silanol groups of hydroxyl nests in zeolites [50]. The bending vibrations of H-O-H for zeolitic H_2_O are displayed by the small peak in the range of 1650–1635 cm^−1^ [45,48].

The fraction of Al, which remained in the solutions at the end of the extraction experiments, was very low, averaging 1.0 ± 0.1% and 3.6 ± 0.3% in experiments that produced GIS- and SOD-zeolites, respectively, (Table 3). Contrary, the fraction of Si remaining in solution at the end of the extraction experiments was relatively high, ranging from 53 to 67% for the GIS-zeolite precipitating experiments and from 68 to 77% for the SOD-zeolite precipitating experiments (Table 3). This evolution of the dissolved Al and Si concentrations is indicative of the high efficiency of the hydrothermal extraction step, i.e., removing the majority of Al from solution and leaving sufficient amounts of Si in solution for the following synthesis step. The relatively higher Si removal in GIS-zeolite forming experiments can be explained by the higher Si content in GIS- vs. SOD-zeolites [40], which is consistent with the FTIR data (Figure 4C,D).

SEM-SEI of solids rich in GIS- (Figure 4E) and SOD-zeolites (Figure 4F) reveal short-prismatic crystals forming bundle-like particle aggregates and nanocrystalline, platy crystals forming globular particle aggregates, respectively. All solids have a high external (surface) porosity and surface roughness, which could arise from an undetermined Na-alumosilicate phase. The crystal edges are partly rounded. No evidence for the presence of unreacted perlite, such as volcanic glass shards, was found, which demonstrates the high efficiency of the hydrothermal conversion process used in the present study. The external SSA of GIS-zeolite and SOD-zeolite are 10.9 m^2^/g and 7.4 m^2^/g, which is ∼2.5 to ∼3.8 times higher compared to the perlite raw material, but lower than most SSA values reported for (modified) natural zeolites [51,52]. We note, however, that the internal SSA of most natural and synthetic zeolites is generally 1–2 orders of magnitude higher, thus providing plenty accessible sites for ion exchange and (ad)sorption reactions.

### 3.2. Hydrothermal Synthesis Step

The XRD patterns reveal that LTA-zeolites are the main precipitates from the extraction solutions treated at 95 °C (Figure 5A) or 65 °C (Figure 5B) for 6 h and 24 h, respectively. Minor amounts of zeolite-X (FAU) as well as GIS- and SOD-zeolites—formed in experiments S1, S3 and S5—are explained either by the initial Si:Al ratio of >1.0, favoring FAU frameworks (S1) and then GIS frameworks (S3) to be formed as the time increases from 6 to 24 h, or by the progressive transformation of LTA- to SOD-zeolite (S5) via interzeolite transformation [53,54] at elevated temperature and higher NaOH concentration in the experimental solution (Table 4). The only exception is the precipitate of experiment S6 because the longer reaction time (24 h), higher temperature (95 °C) and higher NaOH content (5 M) favored a SOD-zeolite to form instead of or from a former LTA-zeolite. Importantly, no crystalline products other than zeolites were identified in all experiments.

The FTIR spectra obtained from the LTA-zeolites (Figure 5C,D) show adsorption in the range of 970–960 cm^−1^, which is related to Si-O-Si and Si-O-Al asymmetrical stretching vibrations for LTA-zeolites [45]. The weak adsorption in the region below 700 cm^−1^ belongs to symmetrical stretching vibrations of the TO_4_ tetrahedra [45]. IR bands in the range from 3800 to 3000 cm^−1^ (data are not shown) and 1650–1640 cm^−1^ correspond to the OH stretching and bending vibrations in zeolitic water, respectively, [45,48].

The fractions of Al and Si remaining in the reacted synthesis solutions at the end of the experiments were extremely low, which in turn demonstrate the high efficiency of the synthesis process (i.e., the removal of Al and Si from solution was determined as 90 ± 9% and 93 ± 6%, on average; Table 4). This near-stoichiometric loss of Al and Si from the solutions is consistent with the ideal Si:Al ratio of ∼1 of zeolite A (LTA) [40].

SEM-SEI of the synthetic materials exhibit a dominantly cube-shaped crystal morphology with smoothed or chamfered edges (and a slightly rough surface) (Figure 5E,F), which is typical for LTA-zeolites formed under hydrothermal conditions [55]. It is evident that the particle size of the solids obtained from experiment S4 is smaller compared to precipitate S2. This is because a higher NaOH concentration favors the nucleation rate of zeolites, which leads to the formation of more but smaller crystals during the subsequent growth state [27]. A lower synthesis temperature generally leads to lower crystal sizes as well [55]. However, in both cases, an overall homogeneous particle size distribution of the LTA-zeolites as well as rounded edges are observed; both size and shape characteristics are beneficial for water treatment applications [29]. The solids obtained from the experiments S3 and S5 additionally contain intergrowths of fine pyramid-shaped crystals (Figure 6A) and thin platy crystals (Figure 6B), respectively, which belong to GIS- and SOD-zeolites. Products others than zeolites have not been identified by SEM imaging. The external SSA of all LTA-zeolites varies from 1.2 to 7.1 m^2^/g.

### 3.3. Ion Sorption Performance

The negative charge of zeolite framework structures and the porous network made of zeolitic cages, cavities and channels of different shape and size (see Figure 1) result in a high chemical affinity towards Me cation uptake and a significant ion exchange capability [56,57]. Herein, the capacity of the synthetic GIS-, SOD- and LTA-zeolites to selectively remove target Me ions was verified in batch experiments. Equilibrium was reached typically within less than 1–2 h for all zeolites investigated, which is in the order expected for micro-/meso-porous (ad)sorbent materials [58]. This observation is consistent with the work by Baldermann et al. [14] where the uptake of Ba^2+^ ions from solution by natural and iron(III)oxide-modified natrolite (NAT) intergrown with stilbite (STI), was studied, which required 30 min to 1 h to reach sorption equilibrium. Similarly, Ibrahim et al. [30] found 30–60 min to be optimal for the removal of heavy metals by zeolite A (LTA) and zeolite X (FAU) synthesized from kaolin. Although the determination of the ion sorption kinetics is beyond the scope of this work, it is widely accepted that Me ion uptake by zeolites is a two-step process, in which (i) initial adsorption takes place in surface-related micropores, followed by (ii) the diffusional transport of the Me ions into the sub-surface zeolite channels, where they occupy the exchangeable sites (e.g., Na^+^, K^+^ and Ca^2+^) within the crystal structure of zeolite minerals following an ion exchange mechanism [14,24,30,59]. In the following sections, the results of the performance tests for some of the most promising (i.e., best-performing) GIS-, SOD- and LTA-zeolites obtained from the hydrothermal conversion of perlite raw material are presented.

#### 3.3.1. Competitive Me Ion Removal from Waste Solution

Waste solutions typically contain distinct pollutants, such as heavy metal ions, and harmless dissolved components of different concentration and type, such as Ca^2+^, Mg^2+^, Na^+^, K^+^, etc., which all have different affinities to sorb onto charged mineral surfaces [60]. In this work, the total removal capacities for Ca^2+^, Mg^2+^, Ba^2+^ and Zn^2+^ ions from an equimolar multi-component waste solution (expressed as CEC in mmol/g) by the synthetic GIS-, SOD- and LTA-zeolites, as well as by the perlite powder and commercial zeolite A, were determined using sorption experiments performed at a liquid:solid ratio of 500:1, at pH ∼6.0 and 25 °C (Figure 7A). It is evident that the perlite has the lowest sorption capacity (0.56 mmol/g) among all materials tested, followed by synthetic SOD-zeolite displaying a moderate CEC (E4: 0.88 mmol/g), and synthetic GIS-zeolite (E2: 2.69 mmol/g), commercial zeolite A (2.73 mmol/g) and synthetic LTA-zeolite (S1–S4: 2.69–2.86 mmol/g), which have the highest CEC values. The highest CEC values are equivalent to a net ion removal capacity of 240 up to 275 mg/g of Me^2+^ for the LTA-zeolites, which illustrates their high efficiency for wastewater treatment [57]. In general, the removal of Me ions occurred in the order Ba^2+^ > Zn^2+^ > Ca^2+^ > Mg^2+^, except for SOD-zeolite and LTA-zeolite S4, where the ion exchange occurred the order Zn^2+^ > Ba^2+^ > Mg^2+^ > Ca^2+^ and Ba^2+^ > Zn^2+^ > Mg^2+^ > Ca^2+^, respectively, (Figure 7A). This selective sorption behavior is explained by the individual structure of the zeolites, the properties of the metal cation such as hydration diameter and hydration energy and the distinct interactions between a certain zeolite and Me ions at the solid–liquid interface [5,56,59]. The porous zeolitic network plays a decisive role in the attraction and repulsion of Me ions, as indicated, e.g., by molecular simulations of selective cation exchange reactions in different zeolite framework structures [61]. Comparing the hydrated radii of the metal cations (Table 5) with the pore openings of the zeolites (Figure 1), it appears that some water must have been separated from the cations during the ion exchange process [5].

Importantly, the sorption capacities measured for the LTA- and GIS-zeolites were considerably high, even under mildly acidic conditions, notwithstanding the fact that most zeolites (i) exhibit an unsuitable performance under such conditions [28] and (ii) preferentially sorb H^+^ ions relative to Me ions in acidic environments [59,62]. The latter is because zeolitic ≡Al-OH and ≡Si-OH groups are protonated under acidic conditions, so that potential binding sites get inactivated by H^+^ ions [14]. However, this effect is negligible under the experimental conditions used in this study. All zeolite materials showed a high resistance under the given acidic conditions, judged by a comparison between the total solid fraction of Si (∼335–680 mg/L) and Al (∼170–330 mg/L) introduced to the batch experiments in the form of zeolite (or perlite) powder and the dissolved fraction of Si (0.2–4 mg/L) and Al (<0.5 mg/L) measured at equilibrium conditions (Figure 7B). In other words, only ∼1–2% of the synthetic zeolites got dissolved in contact with the multi-component waste solution.

**Table 5 materials-15-04474-t005:** Radii of hydration shells, ionic radii, and hydration energies for selected Me ions. The hydrated radius was obtained from [63]; the ionic radius and the hydration energies were obtained from [64].

Cation	Hydrated Radius	Ionic Radius	Hydration Energy
	(Å)	(Å)	(kJ/mol)
Na^+^	3.58	1.02	−365
K^+^	3.31	1.38	−295
Ca^2+^	4.12	1.00	−1505
Mg^2+^	4.28	0.72	−1803
Ba^2+^	4.04	1.36	−1250
Zn^2+^	4.30	0.75	−1955

#### 3.3.2. Ca and Mg Ion Removal for Water Softening

The specific removal capacities of Ca^2+^ and Mg^2+^ ions by the different synthetic zeolites were determined using ion sorption experiments through the reaction with Graz tap water (2.1 mmol/L Ca^2+^, 0.6 mmol/L Mg^2+^, 0.3 mmol/L Na^+^ and 0.03 mmol/L K^+^, at pH 7.8 and 25 °C) using a liquid:solid ratio of 500:1 (Figure 8). The cation uptake from solution followed in the order Ca^2+^ > Mg^2+^ >> Na^+^ and K^+^ for all materials tested and the removal efficiency increased in the order perlite << SOD-zeolite (E4) < GIS-zeolite (E2) < Na-A (reference) and LTA-zeolites (S1–S4). The removal capacities measured for the LTA-zeolites vary from 0.99 up to 1.27 mmol/g (Figure 8A), which is equivalent to CEC values of ∼39–51 mg/g and corresponding removal efficiencies for Ca^2+^ and Mg^2+^ ions of 82–99.8% and 63–93%, respectively, (Figure 8B). These ion removal capacities are in the same order of magnitude compared to a suite of adsorbents used for ion removal (Ca^2+^, Mg^2+^ and Na^+^) from aqueous solutions [24,26,57], although the reported CEC values do not necessarily indicate the maximum sorption capacities of the synthesized zeolites. The K^+^ concentrations remained at the same level or increased slightly in all experiments, whereas the Na^+^ concentrations increased significantly in the equilibrated solutions. This suggests that Na^+^ and K^+^ ions were liberated into the solution via ion exchange reactions that took place in the surface-related micro-pores and exchangeable sites in zeolites [14]. In essence, especially the synthetic LTA-zeolites perform similar or even better than commercial zeolite A, rendering a potential application as water softening agent possible.

### 3.4. Comparison of Adsorbents

The metal ion uptake depends, among other factors, on several parameters such as the nature of the adsorbate solution, the type of zeolite, the adsorbent dosage, the pH of the system, contact time and temperature, so that the comparison between different studies is not always straightforward [14]. Furthermore, adsorption in a multi-ion system differs from single-component systems because the interactions between solutes and possible competing adsorption behaviors (e.g., repulsing effects of cations on each other) are involved [5,61]. Nevertheless, Table 6 shows a comparison between the adsorption capacities for aqueous Me ions obtained in this study and those reported in the literature using comparable conditions and zeolitic adsorbents. The LTA-zeolite synthesized in this work (e.g., S2) shows a significantly higher adsorption capacity for Ba^2+^ compared to zeolite composite materials synthesized from fly ash [65] and higher Zn^2+^ adsorption than the zeolite A produced from kaolin [30] or fly ash [33] and a natural clinoptilolite [66]. On the contrary, commercial zeolite 4A displays a higher Ba^2+^ uptake [5] and the zeolite A synthesized from fly ash by Izidoro et al. [67] has a higher Zn^2+^ adsorption capacity. The sorption capacities for Ca^2+^ for the LTA-zeolite synthesized from alum. sludge [26] and Ca^2+^ and Mg^2+^ for the mesoporous LTA-zeolite [24] are slightly higher than those of S2, however, the total adsorption percentages for Ca^2+^ and Mg^2+^ for S2 are as high as 99.8% and 93.3%, respectively, which makes a direct comparison difficult. The Ba^2+^ uptake of the GIS-zeolite (E2) is second only to the commercial zeolite 4A [5] and has a similar Zn^2+^ uptake compared to the other materials, while the SOD-zeolite (E4) has the lowest Ba^2+^ uptake of all the materials in Table 6, the Zn^2+^ adsorption capacity levels that of the other materials. An exception for the Zn^2+^ uptake for both the GIS- and SOD-zeolites is again the zeolite A synthesized from fly ash by Izidoro et al. [67].

## 4. Conclusions

The perlite powder studied here has a high amount of reactive silica (>70 wt.%), which is beneficial for the synthesis of high quality and high quantity zeolite A (LTA) via a two step reaction process. In the first reaction step, impurities (quartz and mica) were removed and GIS-/SOD- zeolites were precipitated at 100 °C and 24 h, leaving a Si-rich solution. Importantly, no evidence for the presence of unreacted perlite was found in all experiments, which demonstrates the high efficiency of the hydrothermal extraction process used in this work, while the extraction with 5 M NaOH results in a higher available dissolved Si for the following synthesis of zeolite A (LTA) (∼53–67% for 2 M vs. ∼68–77% for 5 M) the considerably better performance of the produced GIS-zeolite and the lower amount of solid NaOH needed as mineralizing agent favors the utilization of 2 M NaOH for the process.

In the second reaction step, the Si-rich solution was mingled with Na-aluminate to synthesize a LTA-zeolite at a high efficiency of 90 ± 9% for Al and 93 ± 6% for Si conversion. A Si:Al ratio of <1 in the precursor alumosilicate gel ensures the formation of pure LTA-zeolite, while a higher ratio leads to the formation of zeolitic impurities. Employing 2 M NaOH, a synthesis duration of 6 h is preferred over 24 h owing to the economic value of shorter synthesis time and the initiating dissolution of metastable zeolite A (LTA) cubes. For the experiments with 5 M NaOH a synthesis temperature of 65 °C is preferred over 95 °C as the higher temperature leads to the formation of an unwanted SOD-zeolite. The high removal capacity of Me^2+^ ions from a multi component waste solution (up to 2.86 mmol/g or 275 mg/g, respectively) for synthesized LTA-zeolites (S1–S4) proves this material to be well suited as adsorbent for wastewater treatment. Furthermore, the best-performing LTA-zeolite removed 99.8% Ca^2+^ and 93.4% Mg^2+^ from the tap water, which demonstrates, together with a favorable shape and size, a high potential for water softening applications. Future work should focus on the optimization of the synthesis conditions and consider effluent (mother liquor) recycling to improve economic viability.

## Figures and Tables

**Figure 1 materials-15-04474-f001:**
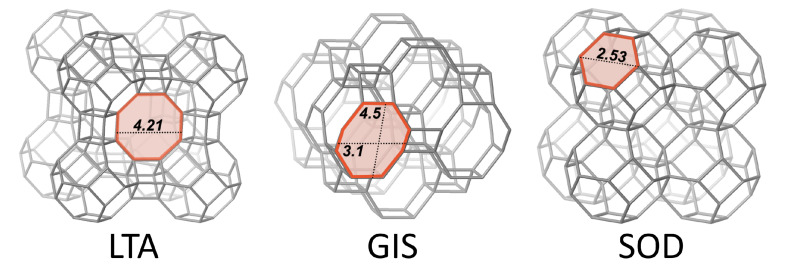
LTA-framework viewed along (100), GIS-framework viewed along (010) and SOD-framework viewed along (010). The respective largest window is highlighted in red and the numbers represent the dimensions in Å (modified after [40]).

**Figure 2 materials-15-04474-f002:**
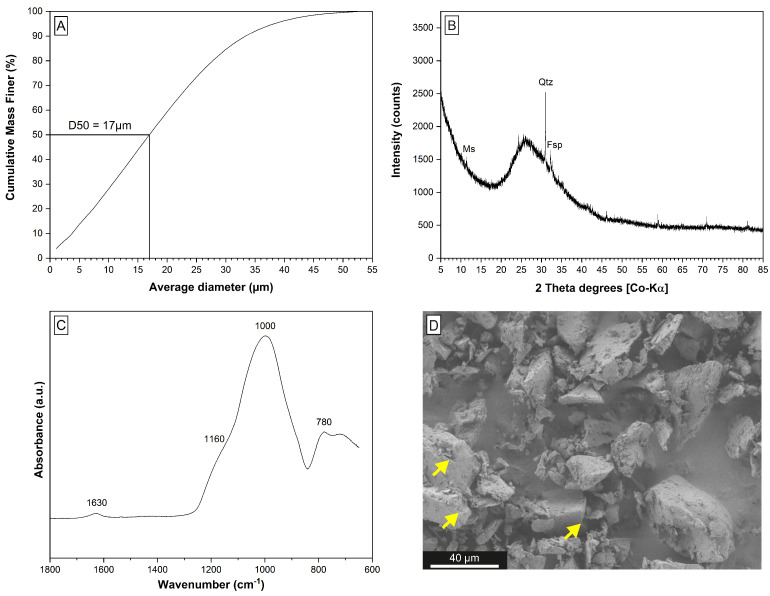
Characterization of the perlite powder used for zeolite synthesis. (**A**) The particle size distribution curve shows the perlite has a grain size of <55 µm and a median diameter (d50) of 17 µm. (**B**,**C**) The XRD pattern and the FTIR spectrum reveal the dominance of an amorphous phase and minor muscovite, quartz, and feldspar in the perlite powder. (**D**) The electron microscope image identifies the amorphous phase as volcanic glass shards, which show conchoidal fractures (marked by the yellow arrows).

**Figure 3 materials-15-04474-f003:**
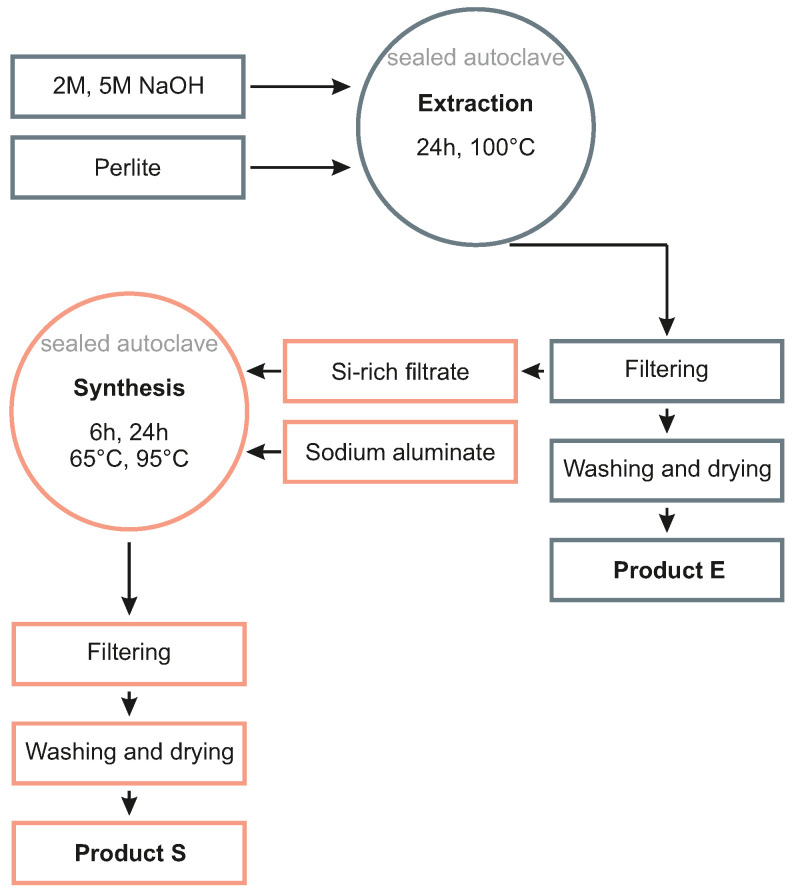
Flow chart showing the two-step synthesis of zeolites via an alkali-hydrothermal method. Target product E: GIS or SOD. Target product S: LTA.

**Figure 4 materials-15-04474-f004:**
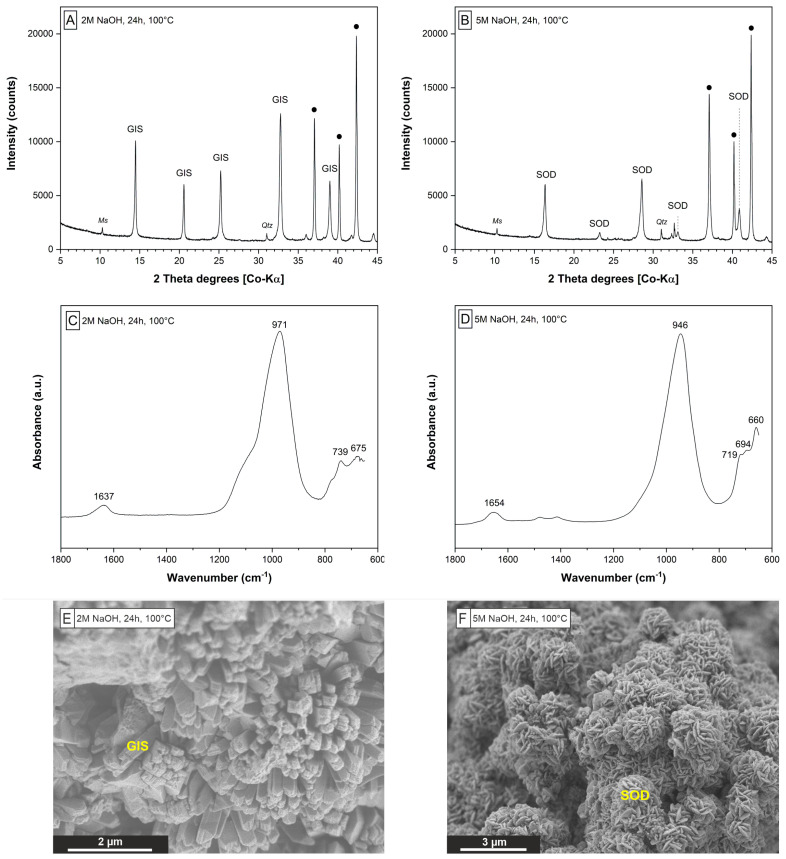
Characterization of zeolites obtained from the hydrothermal extraction step 1. XRD patterns (**A**,**B**), FTIR spectra (**C**,**D**) and SEM-SEI (**E**,**F**) of solids obtained from experiments E2 (left column) and E4 (right column) show GIS and SOD are the main zeolites precipitated. •—marks the peaks derived from the internal zincite standard.

**Figure 5 materials-15-04474-f005:**
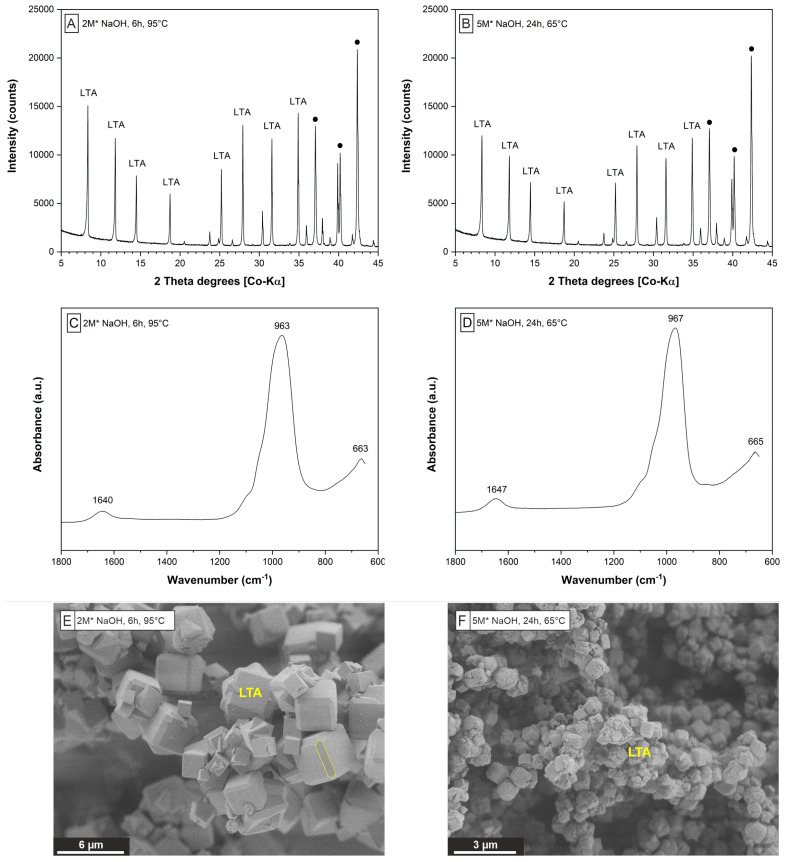
Characterization of zeolites obtained from the hydrothermal synthesis step 2. XRD patterns (**A**,**B**), FTIR spectra (**C**,**D**) and SEM-SEI (**E**,**F**) of solids obtained from experiments S2 and S4 reveal the formation of LTA-zeolites with chamfered edges (highlighted in **E**). •—marks the peaks derived from the internal zincite standard.

**Figure 6 materials-15-04474-f006:**
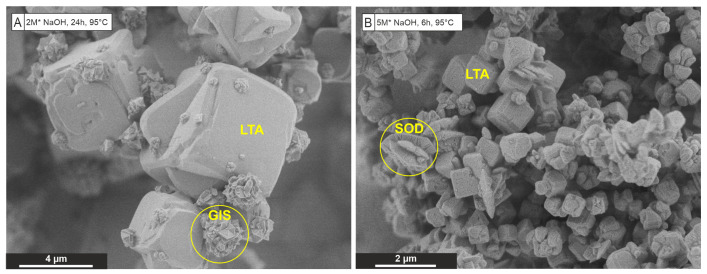
Characterization of zeolites obtained from the hydrothermal synthesis step. SEM-SEI of solids obtained from experiments S3 (**A**) and S5 (**B**) reveal the dominance of LTA-zeolites with minor GIS- and SOD-zeolites (highlighted with yellow color).

**Figure 7 materials-15-04474-f007:**
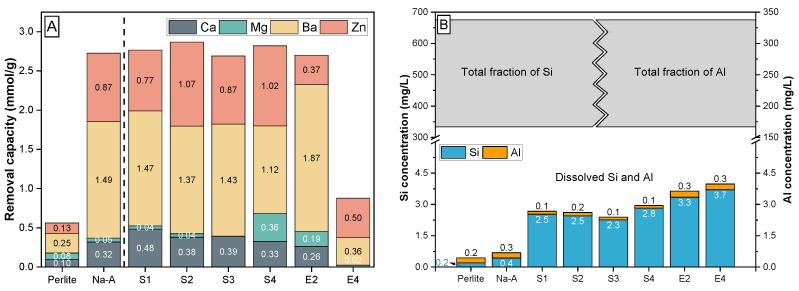
(**A**) Removal capacity for Ca^2+^, Mg^2+^, Ba^2+^ and Zn^2+^ ions by synthetic GIS- (E2), SOD- (E4) and LTA-zeolites (S1–S4) from a multi-component waste solution (∼4 mmol/L of Me^2+^; pH ∼ 6; 25 °C). Perlite powder and commercial zeolite A (Na-A) are included as a reference. (**B**) The low concentrations of dissolved Al and Si measured at ion exchange equilibrium with the acidic multi-component waste solution, relative to the high amounts of solid perlite and zeolite-related Al and Si added to system (grey box), indicate the strong chemical resistance of the synthesized zeolites. In total, less than 2% of the materials was dissolved.

**Figure 8 materials-15-04474-f008:**
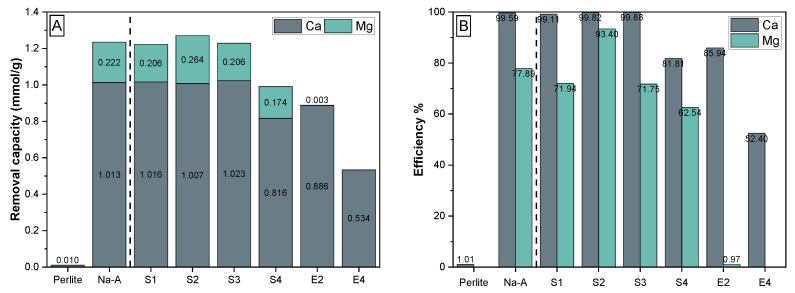
(**A**) Removal capacity for Ca^2+^ and Mg^2+^ ions by synthetic GIS- (E1), SOD- (E2) and LTA-zeolites (S1–S4) from Graz tap water. Perlite powder and commercial zeolite A (Na-A) are included as a reference. (**B**) Removal efficiencies for Ca^2+^ and Mg^2+^ ions measured at ion exchange equilibrium with Graz tap water, demonstrating the applicability of the zeolites for water softening purposes.

**Table 1 materials-15-04474-t001:** Overview of selected zeolitic framework structures and compositions relevant for this study. Data for composite building units (CBU’s), type materials and framework compositions was obtained from [40].

Framework	CBU’s	Type Material	This Work	Framework Composition
LTA	*d4r, sod, lta*	Linde Type A	Zeolite A	[Al_12_Si_12_O_48_]_8_
GIS	*gis*	Gismondine	Na-P1	[Al_6_Si_10_O_32_]
SOD	*sod*	Sodalite	Hydro(xy)sodalite	[Al_6_Si_6_O_24_]

**Table 2 materials-15-04474-t002:** Chemical composition of the perlite raw material used for zeolite synthesis.

Na_2_O	K_2_O	CaO	MgO	Fe_2_O_3_	Al_2_O_3_	SiO_2_	P_2_O_5_	LOI	SUM
3.4	3.9	0.7	<0.1	1.4	14.1	72.2	0.1	4.2	99.9

**Table 3 materials-15-04474-t003:** Experimental conditions used for the hydrothermal synthesis of GIS- and SOD-zeolites from perlite powder (step 1). Si_aq_ and Al_aq_ denote the fractions of Si and Al remaining in the reacted solution at the end of the first extraction step. The mineralogical composition of the solids (product E) is indicated on the right. Quartz (Qtz) and muscovite (Ms) are impurities that are inherited from the perlite raw material.

Sample	c(NaOH)	Si:Al	Na:Al	Time	Temp.	Si_aq_	Al_aq_	Main	Silicates
	(mol/L)	(mol)	(mol)	(h)	(°C)	%	%	Phase	
E1	2	4.3	7.6	24	100	60.8	1.0	GIS	Qtz,Ms
E2	2	4.3	7.5	24	100	53.4	0.9	GIS	Qtz,Ms
E3	2	4.3	7.6	24	100	67.4	1.0	GIS	Qtz,Ms
E4	5	4.3	18.3	24	100	73.3	3.5	SOD	Qtz,Ms
E5	5	4.3	18.4	24	100	67.5	3.3	SOD	Qtz,Ms
E6	5	4.3	18.4	24	100	77.2	3.9	SOD	Qtz,Ms

**Table 4 materials-15-04474-t004:** Experimental conditions used for the hydrothermal synthesis of LTA-zeolites (step 2). The SAR was adjusted to ∼1.0 by the addition of solid NaAlO_2_. ε_Si_ and ε_Al_ indicate the removal efficiency of Si and Al from solution after the synthesis step. The mineralogical composition of the solids (product S) is indicated on the right, documenting LTA to be the dominating zeolite.

Sample	Si:Al	Na:Al	Time	Temp.	$\varepsilon_{Si}$	$\varepsilon_{Al}$	Main	Impurities
	(mol)	(mol)	(h)	(°C)	%	%	Phase	
S1	1.2	4.4	6	95	87.4	98.7	LTA	FAU
S2	0.7	3.1	6	95	99.4	79.6	LTA	-
S3	1.1	4.2	24	95	96.7	97.0	LTA	GIS
S4	0.9	5.7	24	65	96.5	94.5	LTA	-
S5	0.8	5.4	6	95	97.0	93.9	LTA	SOD
S6	1.0	6.2	24	95	93.9	98.0	SOD	-

**Table 6 materials-15-04474-t006:** Comparison of the sorption capacities of various materials for the removal of Me ions from solution under comparable operational conditions. *—indicates Q-values derived from competitive/multicomponent ion sorption experiments.

Adsorbent	Q (mg/g)	Operating Conditions					Ref.
		C_i_ (mg/L)	Time (h)	Temp. (°C)	(Optimum) pH	Dosage (g/mL)	
commercial zeolite 4A	Ba: 309.0 Sr: 205.0	41–3433 26–2191	14	25	8.2–11.1	0.0075	[5]
mesoporous zeolite LTA	Ca: 55.7 Mg: 9.2	160 97	0.7	35	–	0.001	[24] *
LTA from alum. sludge	Ca: 65.5	99	72	30	7.2	0.003	[26] *
zeolite A from kaolin	Cd: 71.4 Cu: 41.3 Pb: 182.3 Ni: 24.7 Zn: 28.8	100–400	0.5–1	25	7.5	0.008	[30]
zeolite A from fly ash	Co: 13.5 Cu: 49.9 Cr: 41.6 Ni: 8.8 Zn: 27.0	300	4	25	3.0	0.005	[33] *
zeolite A from fly ash	Cd: 185.1 Zn: 219.6	1121–3372 654–1961	24	RT	6.6–6.8	0.01	[67]
Na-P1 from fly ash	Ni: 20.1 Zn: 32.6	10–200	6	22	6.0	0.0025	[66]
Hydroxysodalite from fly ash	Pb: 153.0	100–1000	6	25	6.0	0.003	[18]
zeolite Z90-4 from ash	Ba: 119.0	50–1000	0.5	50	4.0–5.0	0.005	[65]
zeolite Z90-15 from ash	Ba: 117.7	50–1000	0.5	50	4.0–5.0	0.005	[65]
natural Clinoptilolite	Ni: 2.0 Zn: 3.5	10–200	6	22	6.0	0.01	[66]
S2 zeolite LTA	Ca: 40.4 Mg: 6.4	84 14	24	25	7.8	0.002	this work *
S2 zeolite LTA	Ca: 15.3 Mg: 1.0 Ba: 188.3 Zn: 70.0	171 103 616 297	24	25	6.0	0.002	this work *
E2 zeolite GIS	Ca: 10.5 Mg: 4.6 Ba: 257.4 Zn: 24.5	171 103 616 297	24	25	6.0	0.002	this work *
E4 zeolite SOD	Ca: 0.0 Mg: 0.5 Ba: 48.9 Zn: 32.6	171 103 616 297	24	25	6.0	0.002	this work *

## Data Availability

All raw data will be made available by request to Florian Painer.

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
