# Peer review of "Synthesis of Zeolites from Fine-Grained Perlite and Their Application as Sorbents"

_materials, 2022, doi:10.3390/ma15134474_

Round 1

Reviewer 1 Report

This is an interesting and fairly well written paper reporting a very solid study concerning possibility of synthesis of commodity zeolites from mineral raw material.

I could recommend publishing it in the present form, provided it were not so long. In my opinion many details concerning experimental work could be easily moved to the Supporting Information. This should help the authors to present the aim of the study, the results and conclusions even more clearly.

One spelling remark: there should be "tap water" in the last but one line of the abstract.

Author Response

1) "This is an interesting and fairly well written paper reporting a very solid study concerning possibility of synthesis of commodity zeolites from mineral raw material. "

We thank the reviewer for the very positive evaluation of our manuscript.

2) "I could recommend publishing it in the present form, provided it were not so long. In my opinion many details concerning experimental work could be easily moved to the Supporting Information. This should help the authors to present the aim of the study, the results and conclusions even more clearly."

We have deleted some general information, especially in the Methods and Analytical sections, in order to shorten the paper and to give more focus on our results and on their interpretation. The total text length of the revised paper has been reduced by 1 - 1.5 pages, and is now in full accordance with the journal`s recommendations. We decided, however, not to move large text parts to the Supporting Information, because the readers should find all essential experimental protocols directly in the main text.

3) "One spelling remark: there should be "tap water" in the last but one line of the abstract."

The text has been changed accordingly.

Reviewer 2 Report

The study described in this manuscript is focused on the use of perlite as material for synthesis of zeolites which can be further used as sorbents of metal ions from waste water. The selected methodology is correct. The results are clearly presented. There is a limited novelty of the study because a few similar research project were already reported in literature. The Authors discuss their results in light of the data already published, so this aspect is well resolved. The language used in the manuscript is correct - a minor comment for line 12: please change "tab" for "tap". I would also consider changing Me for M as Me is usually used as abbreviation for methanol, but this remark is not critical.

I would recommend the article to be published as is.

Author Response

1) "The study described in this manuscript is focused on the use of perlite as material for synthesis of zeolites which can be further used as sorbents of metal ions from waste water. The selected methodology is correct. The results are clearly presented. There is a limited novelty of the study because a few similar research project were already reported in literature. The Authors discuss their results in light of the data already published, so this aspect is well resolved. (…) I would recommend the article to be published as is."

We thank the reviewer for the very positive evaluation of our manuscript.

2) "The language used in the manuscript is correct - a minor comment for line 12: please change "tab" for "tap"."

The text has been changed accordingly.

3) "I would also consider changing Me for M as Me is usually used as abbreviation for methanol, but this remark is not critical."

We fully agree. However, we decided not to change “Me” for “M”, because “M”, as a unit, stands for molarity, i.e., we use the unit “M” in the Methods section (and elsewhere) to define molar concentrations of e.g., our extraction/synthesis solutions. For this reason, we continue to use “Me” for metal ions and “M” to define molar concentrations.

Reviewer 3 Report

This article presents very dense and interesting data which producing excellent results about synthetic zeolite A from perlite processed. However some concern need to be clarified and revised from the author:

1. In background, low cost become one of consideration in paragraph 3 (line 43-57), particularly to synthesis LTA-zeolite. However, I can not find in your background part the comparable results when using the natural material sorbent directly or in simple way treatment to be applied in water treatment as you mentioned in paragraph 2 (line 35-42). I believe that the natural sorbent (or with simple way treatment) i.e natural zeolite must have low cost than the synthetic one with the complex processing; also has abundant stock for some type of them. 

2. Based on the LTA-zeolite characterization resulted from this synthesis, the SSA is one of the important targeted result. In line 87, the perlite has SSA of 2.9 m2/g; then at line 283, the synthetic zeolite has SSA of 10.9 m2/g and 7.4 m2/g, so it increased more than 2.5 times from the raw material. It is good indication of sorbent characterization, particularly to support physical sorption ability. However in some literature, I found that there are many natural zeolites which have higher SSA (even without any treatment) which have possibility have much better performance with low cost process to provide it. Please explain in your paper, what is the main characteristics target and why it have important impact (the main characteristic of zeolite usually about SSA, Si/Al ratio, crystal type, Pore Size Distribution, etc). Some reference which mention about the SSA of natural zeolite and their change in various treatment such as:

Wahono, Satriyo Krido, et al. "Physico-chemical modification of natural mordenite-clinoptilolite zeolites and their enhanced CO2 adsorption capacity." Microporous and Mesoporous Materials 294 (2020): 109871.

Wahono, Satriyo Krido, et al. "Multi-stage dealumination for characteristic engineering of mordenite-clinoptilolite natural zeolite." AIP Conference Proceedings. Vol. 2085. No. 1. AIP Publishing LLC, 2019.

Dziedzicka, Anna, Bogdan Sulikowski, and MaÅ‚gorzata Ruggiero-MikoÅ‚ajczyk. "Catalytic and physicochemical properties of modified natural clinoptilolite." Catalysis Today 259 (2016): 50-58.

Mierczynski, Pawel, et al. "Biodiesel production on monometallic pt, pd, ru, and ag catalysts supported on natural zeolite." Materials 14.1 (2020): 48.

3. Which one do you want to provide for this material, the chemisorption or physisorption as the dominant ability for your application?

4. In line 240, the preconditioning for zeolite is heating at 105 C, but in some reference for zeolite material the preconditioning or outgassing for BET analysis need to be conducted above 200 C (ideally at 350-400 C) for zeolite LTA4A (Figini-Albisetti, Alessandro, et al. "Effect of outgassing temperature on the performance of porous materials." Applied Surface Science 256.17 (2010): 5182-5186) and treated A5 natural zeolite (Wahono, Satriyo K., et al. "Amine-functionalized natural zeolites prepared through plasma polymerization for enhanced carbon dioxide adsorption." Plasma Processes and Polymers 18.8 (2021): 2100028). Particularly for the determination of the zeolite synthetic product, it will provide less accurate data of SSA.

Author Response

"This article presents very dense and interesting data which producing excellent results about synthetic zeolite A from perlite processed. However some concern need to be clarified and revised from the author:"

We thank the reviewer for the overall very positive evaluation of our manuscript.

1) "In background, low cost become one of consideration in paragraph 3 (line 43-57), particularly to synthesis LTA-zeolite. However, I can not find in your background part the comparable results when using the natural material sorbent directly or in simple way treatment to be applied in water treatment as you mentioned in paragraph 2 (line 35-42). I believe that the natural sorbent (or with simple way treatment) i.e natural zeolite must have low cost than the synthetic one with the complex processing; also has abundant stock for some type of them."

We completely agree with the reviewer. We have provided the following explanation for this frequent economic challenge: “In essence, while the use of natural zeolites in water treatment is cost-efficient, the coefficient of performance is often lower compared to the application of synthetic, high-performance LTA-zeolites at industrial scale. For this reason, the production of LTA-zeolites from industrial raw materials can serve as a ‘green solution’, as this approach reduces the demand for primary raw materials and minimizes the costs for residues disposal (Schlögl et al., 2022)”

2) "Based on the LTA-zeolite characterization resulted from this synthesis, the SSA is one of the important targeted result. In line 87, the perlite has SSA of 2.9 m2/g; then at line 283, the synthetic zeolite has SSA of 10.9 m2/g and 7.4 m2/g, so it increased more than 2.5 times from the raw material. It is good indication of sorbent characterization, particularly to support physical sorption ability. However in some literature, I found that there are many natural zeolites which have higher SSA (even without any treatment) which have possibility have much better performance with low cost process to provide it. Please explain in your paper, what is the main characteristics target and why it have important impact (the main characteristic of zeolite usually about SSA, Si/Al ratio, crystal type, Pore Size Distribution, etc). Some reference which mention about the SSA of natural zeolite and their change in various treatment such as:

- Wahono, Satriyo Krido, et al. "Physico-chemical modification of natural mordenite-clinoptilolite zeolites and their enhanced CO2 adsorption capacity." Microporous and Mesoporous Materials 294 (2020): 109871.

- Wahono, Satriyo Krido, et al. "Multi-stage dealumination for characteristic engineering of mordenite-clinoptilolite natural zeolite." AIP Conference Proceedings. Vol. 2085. No. 1. AIP Publishing LLC, 2019.

- Dziedzicka, Anna, Bogdan Sulikowski, and Małgorzata Ruggiero-Mikołajczyk. "Catalytic and physicochemical properties of modified natural clinoptilolite." Catalysis Today 259 (2016): 50-58.

- Mierczynski, Pawel, et al. "Biodiesel production on monometallic pt, pd, ru, and ag catalysts supported on natural zeolite." Materials 14.1 (2020): 48. "

We fully agree with the reviewer and acknowledge this important comment. Besides key parameters like Si/Al ratio, crystal type, pore size distribution, purity etc., zeolite structures are characterized by an internal SSA (e.g., mainly predefined by the zeolite type and network structure) and an external SSA (e.g., mainly defined by the zeolite form, size and roughness). To fully excess and determine the internal SSA via ‘conventional’ BET measurements (heat treatment at 105 °C), higher preconditioning (up to 400 °C) is required, as correctly indicated by the reviewer. However, DTA (data not shown) suggest a beginning modification of the LTA-zeolites via recrystallization already at a temperature of <300 °C. We therefore decided to analyze only the external SSA of our zeolites, but note that this property is unfortunately not fully representative of the total metal ion removal potential, which is better represented by the ion-accessible internal SSA. We have provided the following explanation for clarity: “The external SSA of GIS-zeolite and SOD-zeolite are 10.9 m2/g and 7.4 m2/g, which is ~2.5 to ~3.8 times higher compared to the perlite raw material, but lower than most SSA values reported for (modified) natural zeolites (Wahono et al., 2020; Dziedzicka et al., 2016). We note, however, that the internal SSA of most natural and synthetic zeolites is generally 1-2 orders of magnitude higher, thus providing plenty accessible sites for ion exchange and (ad)sorption reactions”. In addition, a homogeneous particle size distribution and rounded edges of the LTA-zeolites are also beneficial for water treatment applications – we mention this on page 10. 

3) "Which one do you want to provide for this material, the chemisorption or physisorption as the dominant ability for your application?"

Metal ion removal from solution by the investigated zeolites predominantly follows an ion exchange mechanism. We have added this information to section 3.3 and 3.3.1. Evidence for this comes from Na and K liberation to the solution, and concurrent uptake of Me ions of the different zeolites.

4) "In line 240, the preconditioning for zeolite is heating at 105 C, but in some reference for zeolite material the preconditioning or outgassing for BET analysis need to be conducted above 200 C (ideally at 350-400 C) for zeolite LTA4A (Figini-Albisetti, Alessandro, et al. "Effect of outgassing temperature on the performance of porous materials." Applied Surface Science 256.17 (2010): 5182-5186) and treated A5 natural zeolite (Wahono, Satriyo K., et al. "Amine-functionalized natural zeolites prepared through plasma polymerization for enhanced carbon dioxide adsorption." Plasma Processes and Polymers 18.8 (2021): 2100028). Particularly for the determination of the zeolite synthetic product, it will provide less accurate data of SSA."

We fully agree with the reviewer again and acknowledge this very important note. However, for the reasons explained above, we provide only data about the external SSA of our zeolites.

Best regards,

The authors

Round 2

Reviewer 3 Report

It is a great improvement from the previous draft and provide more clarity to the readers for understanding the contains of this paper. Hopefully, it provides benefits great development for science and industrial applications.